# Towards Intraoperative Quantification of Atrial Fibrosis Using Light-Scattering Spectroscopy and Convolutional Neural Networks

**DOI:** 10.3390/s21186033

**Published:** 2021-09-09

**Authors:** Nathan J. Knighton, Brian K. Cottle, Bailey E. B. Kelson, Robert W. Hitchcock, Frank B. Sachse

**Affiliations:** 1Department of Biomedical Engineering, University of Utah, 36 South Wasatch Drive, Salt Lake City, UT 84112, USA; nate.knighton@gmail.com (N.J.K.); brian.cottle@utah.edu (B.K.C.); baileykelson@gmail.com (B.E.B.K.); robert.hitchcock@utah.edu (R.W.H.); 2Nora Eccles Harrison Cardiovascular Research and Training Institute, University of Utah, 95 S 2000 E, Salt Lake City, UT 84112, USA

**Keywords:** atrial fibrosis, ablation lesions, light-scattering spectroscopy, optical biopsy, convolutional neural network

## Abstract

Light-scattering spectroscopy (LSS) is an established optical approach for characterization of biological tissues. Here, we investigated the capabilities of LSS and convolutional neural networks (CNNs) to quantitatively characterize the composition and arrangement of cardiac tissues. We assembled tissue constructs from fixed myocardium and the aortic wall with a thickness similar to that of the atrial free wall. The aortic sections represented fibrotic tissue. Depth, volume fraction, and arrangement of these fibrotic insets were varied. We gathered spectra with wavelengths from 500–1100 nm from the constructs at multiple locations relative to a light source. We used single and combinations of two spectra for training of CNNs. With independently measured spectra, we assessed the accuracy of the CNNs for the classification of tissue constructs from single spectra and combined spectra. Combined spectra, including the spectra from fibers distal from the illumination fiber, typically yielded the highest accuracy. The maximal classification accuracy of the depth detection, volume fraction, and permutated arrangements was (mean ± standard deviation (stddev)) 88.97 ± 2.49%, 76.33 ± 1.51%, and 84.25 ± 1.88%, respectively. Our studies demonstrate the reliability of quantitative characterization of tissue composition and arrangements using a combination of LSS and CNNs. The potential clinical applications of the developed approach include intraoperative quantification and mapping of atrial fibrosis, as well as the assessment of ablation lesions.

## 1. Introduction

Optical biopsy has the potential to revolutionize medical diagnostics by quantifying interactions of light with tissue. An optical biopsy eliminates many of the limitations of a conventional biopsy based on tissue excision. These limitations include tissue damage, infections, pain, and delayed analysis. Diseases such as cancers, infections, fibrosis, and organ rejection have been diagnosed or studied in vivo using optical biopsy. An optical biopsy is often based on imaging modalities such as multi-photon microscopy, optical coherence tomography, and fiber-optics confocal microscopy (FCM) [1,2,3]. These modalities differ in their technical complexity, cost, and suitability for use in the various clinical environments, including the operating room. In addition, these different approaches provide various imaging depths and spatiotemporal resolutions which may limit their clinical applications.

An alternative to these imaging modalities is optical spectroscopy [4]. For example, light-scattering spectroscopy (LSS) has been used for the optical biopsy of various tissues in the human body [3,5,6,7]. LSS is based primarily on the scattering and absorption of photons with tissue constituents [8,9]. In many biological tissues, including the myocardium and aorta, the probability of scattering is much larger than the probability of absorption for light in the visible and near-infrared wavelength. Major scatterers in these tissues are cell nuclei, mitochondria, and collagen and their scattering properties are wavelength and size-dependent [4]. LSS uses light in a wavelength range to illuminate the tissues. Light scattered by tissue components towards a receiver is collected as a spectrum. Thus, LSS provides information corresponding to the tissue constituents, which allows the assessment of their density and size. Light scattering can be particularly pronounced in the surface layers of the tissue. Current clinical applications of LSS, such as early detection of cancers [5,6], benefited from isolating the strong single scattering signal at tissue surfaces, in particular, from the nuclei of superficial cells. In contrast, the developments of LSS for volumetric assessment of tissues have only been sparsely reported.

While most LSS systems require only basic optical components, analysis and interpretation of the acquired spectra are often based on simplistic models and time-consuming simulations of light-tissue interactions, for instance, Monte Carlo methods. Also, analyses were developed focusing on specific scatterers, e.g., superficial nuclei. A general framework for analyses of the spectra from cardiac tissues has not been introduced. In many biomedical fields, machine learning has been proposed and is increasingly applied to overcome similar challenges. Various approaches of machine learning have been applied, e.g., for the detection of bacterial serotype based on optical scattering, diagnosis of ischemic heart disease from electrocardiograms, and human joint classification from diffuse reflectance spectroscopy [10,11,12]. In our prior work, we applied a machine learning approach, i.e., a convolutional neural network (CNN), to detect fibrosis in images from catheterized FCM of the beating heart in situ in an animal model [13].

Here, we investigated LSS as an approach for the characterization and discrimination of cardiac tissues and their remodeling in disease. We implemented a system for broad-spectrum LSS having a single illumination and multiple collection fibers (Figure 1). The spectra were analyzed using an approach that included a one-dimensional CNN. We applied the system for measurements on tissue constructs comprising sections of the myocardium and aortic wall (Figure 2). The tissue constructs had a thickness similar to the atrial free wall. The aorta wall served as a substitute for tissue with a high collagen content, such as fibrotic atrial myocardium. Other tissues in the heart with a high collagen content include scar and connective tissue. Spectra measured from the tissue constructs were used to train and test CNNs. We evaluated the ability of the CNNs to identify fibrotic insets placed at varying depths in the myocardium, volume fractions of myocardial and fibrotic tissues, and permutated arrangements of these tissues. Using CNNs, we identified the optimal arrangement of collection fibers for cardiac tissue characterization. The studies provide important insights into the potential applications of LSS for the quantification and mapping of fibrosis in atrial tissues.

## 2. Materials and Methods

### 2.1. Tissue Preparations and Experiments

All animal usage was approved by the Institutional Animal Care and Use Committee, University of Utah. Adult female rats were anesthetized and then euthanized by exsanguination. The hearts were excised and Langendorff perfused with Tyrode’s solution. We also used tissues from formalin-fixed adult canine hearts. All of the research was performed in accordance with the relevant guidelines and regulations.

### 2.2. Explorative Study

We explored the ability of the LSS system to detect the differences in spectra from live rat myocardium and aorta of four female rats. The illumination fiber was placed on the epicardial surface of the left ventricle. A collection fiber was mounted in a custom holder attached to a 3-axis micromanipulator (MP-285, Sutter Instrument Company, Novato, CA, USA). The manipulator was used to acquire spectra at 100 µm intervals outward from the illumination fiber. Every other measurement was taken in an orthogonal direction relative to the preceding one to explore any effects of tissue anisotropy. After measurements, the heart was retrogradely perfused using 2% paraformaldehyde in Tyrode’s solution for fixation. The spectra were gathered at 1-min intervals over 15 min during fixation. We calculated the Pearson correlation coefficient (R_r_) to compare the spectra from the live and fixed tissues [14]:(1)$$R_{r}\left(x,y\right)=\frac{\sum_{i=1}^{n}\left(x_{i}-\;\bar{x}\right)\left(y_{i}-\;\bar{y}\right)}{\sqrt{\sum_{i=1}^{n}{\left(x_{i}-\;\bar{x}\right)}^{2}\sum_{j=1}^{n}{\left(y_{i}-\;\bar{y}\right)}^{2}}}$$
for the spectra x and y with means $\bar{x}$ and $\bar{y}$, and the number n of data points within a spectrum.

We assessed our LSS system by measuring the SNR of 100 spectra sampled at 1, 10, 20, 27, 30, 33, 40, and 50 ms using both the calibration standard and myocardial tissue. The variation of the sampling rate served as an analogue for the exploration of varied light source intensity on the signal quality.

### 2.3. Normalization and Calibration of Spectra

The dark noise and background in the spectra were corrected by the spectrometer internal software. The spectra were normalized to their mean intensity. The calibration of the spectra of tissues applied a spectrum measured from a white diffuse reflectance standard reference with a 99% reflectance factor (Spectralon^®^, Labsphere, Inc., North Sutton, NH, USA).

### 2.4. Experimental Design

In addition to exploring our approach in a rat, we adapted the experimental approach for use with fixed canine tissue constructs created by placing thin tissue sections of myocardium or aortic tissue on top of each other. Tissue samples from the right ventricular free wall of a fixed canine heart and ascending aorta were obtained using a biopsy punch with a diameter of 5 mm. Tissue sections with a thickness of 200 µm and diameter of 5 mm were created using a cryotome (Leica CM1950, Leica Biosystems Inc, Wetzlar, Germany) and stored in phosphate-buffered saline (PBS). The construct height was limited to a maximum of 8 sections or 1.6 mm in total height, which is within the reported region-specific thicknesses of the atrial wall of a healthy adult human [15]. Constructs were built in a PBS-filled bin and bottom-up on a black open-cell foam pad to mitigate the potential effects of light reflection and scattering from structures outside of the constructs. When moving the stored tissue sections, we assured that they did not exhibit air pockets. The tissue constructs were built while immersed in PBS and kept immersed during spectroscopy to prevent the capture or retention of air pockets. The tissue arrangement within constructs was dependent on the classification objective.

In these studies, we used constructs that were assembled from sections of ventricular myocardium and aorta from a canine. The constructs were named by their composition of sections, and read left-to-right with the leftmost value being the top layer of the constructs. Myocardial and aortic sections were identified by “0” and “1”, respectively. Thus, ‘00000000′ is a construct of only myocardium, ‘11110000′ is a construct of aortic tissue in the top four layers with myocardium in the bottom four layers, and ‘11111111′ is a construct of only aortic tissue.

We built a spectroscopic probe integrating an illumination fiber with 5 collection fibers to simplify the gathering of multiple spectra from the tissue constructs (Appendix A). The collecting fibers were labeled R1 through R5 and corresponded to distances from the center of the illumination fiber of 210, 345, 480, 615, and 750 µm, respectively. The probe was placed at different locations and orientations on the construct for each measurement. From each construct, we gathered 10 spectra per collection fiber. Each spectrum was measured for 30 ms. The tissue constructs were rebuilt with new sections after 10 measurements. At least 20 measurements were performed for each arrangement. The measured spectra were stored and are publicly available at [16].

### 2.5. CNN Design

We employed the Keras neural networks application programming interface (API) (2.3.1) in conjunction with Tensorflow (2.2.0) to develop CNNs for classifying the tissue constructs from the measured spectra. The CNN topology is described in Table 1 [17,18,19,20]. The same topology was used for all of the experimental assessments. For each assessment, we trained 10 CNNs on the measured spectra. We used the stochastic adaptive moment estimation (ADAM) optimizer with a learning rate of 0.0001 [21]. We separated 30% of the training dataset to be used to calculate the validation loss and accuracy values during the training process. Our testing scheme utilized subject-wise hold-out cross-validation wherein all the spectra from a single construct of each class were withheld for testing. The batch sizes for each dataset were selected to include the entire training dataset after the removal of the validation and testing datasets. The loss values for the multi-class classifier were calculated by a cross-entropy loss followed by a softmax activation. The loss values for the binary classifier were calculated using a cross-entropy loss followed by a sigmoid activation. The training was stopped when the training loss did not decrease for at least 400 epochs or after 1000 total epochs. A small number of networks with very low accuracy values were excluded from the analyses.

Each CNN was then evaluated for classification accuracy using the test dataset. Classification accuracy was calculated as:(2)$$\mathrm{Accuracy}=\frac{n_{\mathrm{correct}}+0.5{\;n}_{\mathrm{similar}}}{n_{\mathrm{total}}}$$
with the number of correct predictions n_correct_, the number of proximal predictions n_similar,_ and the total number of predictions n_total_. Due to the partially high similarity of the tissue constructs underlying our classes used in our experimental studies, the proximal predictions were counted as 50% correct predictions. We defined a proximal prediction as a prediction that differed from the correct prediction by the minimal difference between classes in the experimental studies. For example, a prediction of the class “11000000” would be considered a proximal prediction for the correct prediction of “00110000” because the difference of the tissue composition between the two underlying classes is the same as the minimum difference between all classes in this experimental study. We present accuracy as the average and standard deviation (stddev) of the accuracy of successfully trained CNNs for each assessment. The accuracies and summary statistics were visualized through confusion matrices.

To validate the network, we trained the CNN to make a binary classification prediction between constructs of all of the myocardium and all of the aorta. We repeated the measurements described in the rat heart for these constructs. We gathered a total of 300 spectra for both classes at five radial distances. As part of this validation, we compared normalized but uncalibrated spectra with normalized and calibrated spectra with respect to testing accuracy.

### 2.6. Assessment of Depth Detection

We tested the sensitivity of our approach to the differences in the depth of an aortic tissue inset within a myocardial construct (Figure 2A). Two sections of aortic tissue were used to create a combined layer of 400 µm thickness. The layer was placed at successively lower positions until all of the positions were measured. We gathered a total of 690 spectra from 6 tissue arrangements at 5 radial distances.

### 2.7. Assessment of Volume Fraction

We studied the ability of our approach to detect volume fractions by varying the relative percentage of fibrotic tissue in a tissue construct by 12.5%, i.e., one 200 µm section, until the entire volume fraction of the construct was fibrotic tissue (Figure 2B). Starting with the second layer, we replaced the top-most layers first to amplify any masking effects the upper layers may have on the lower layers. The collected dataset consisted of 1350 spectra from 5 radial distances in 9 tissue arrangements.

### 2.8. Assessment of Permutated Tissue Arrangements

We further explored the approach with arrangements permutating myocardial and aortic sections where the total volume fraction of each tissue type was 50%. The constructs in Figure 2C illustrate the explored arrangements. We measured 900 spectra at 5 radial distances from 6 tissue arrangements.

### 2.9. Analysis of Combinations of Collection Fibers

We explored the effects of combining the spectra from different collection fibers to test whether fiber combinations improved the CNN classification accuracy over single fibers. The spectra from various pairs of collection fibers formed combined spectra (R1R2, R1R3, …, R4R5). We established a fiber combination hierarchy by identifying the most accurate combination for depth, volume fraction, and arrangement cases.

## 3. Results

### 3.1. Development of a Broad-Spectrum LSS System

Figure 1 illustrates the developed LSS system and experimental setup to measure and analyze light-scattering spectra from cardiac tissues. A stabilized tungsten-halogen broad-spectrum light source (SLS201L/M, Thorlabs, Newton, NJ, USA) provided illumination in the 360–2500 nm range through a single fiber with a core diameter of 200 µm (FG200LCC, Thorlabs). The light source provides a coupled optical power of 10 mW, which was used for all experiments. Collection fibers of 105 µm core diameter (FG105LCA Thorlabs) gathered the spectra, which were measured using a Czerny–Turner type CCD spectrometer (CCS175/M, Thorlabs) having a wavelength detection range of 500–1100 nm and a full width at half maximum spectral accuracy of less than 0.6 nm at 633 nm.

### 3.2. Spectroscopy in Living and Fixed Cardiovascular Tissues

We measured the spectra from the myocardium and aorta of Langendorff-perfused rat hearts (Figure 3A). Consistent with the previous studies [22,23], we found differences in the spectra, such as increased intensities in the aorta vs. myocardium. The larger density of collagen and cell nuclei in the aorta vs. myocardium is responsible for these differences of the spectra as explained in previous research [23]. The correlation of the spectra from the two tissues was moderate (R_r_ = 0.87 ± 0.08). The correlation between the myocardial spectra before and after the tissue fixation was high (R_r_ = 0.98 ± 0.02) (Figure 3B). We next characterized the anisotropy of the spectra by comparing the measurements along one direction with the measurements along an orthogonal direction. The raw spectra from the measurements at orthogonal directions exhibited a high correlation in live aortic tissue (R_r_ = 0.99 ± 0.01) and live myocardium (R_r_ = 0.95 ± 0.03). The fixed tissue also exhibited a high orthogonal correlation in aortic (R_r_ = 0.98 ± 0.02) and myocardial (R_r_ = 0.92 ± 0.03) tissue. Based on these findings, we used fixed tissues and included the spectra from sites at the orthogonal directions in both the training and test datasets in subsequent studies.

We assessed our LSS system by measuring the signal-to-noise ratio (SNR). It increased approximately linearly with a sampling rating for both the calibration standard (Figure 3C) and myocardial tissue stacks (Figure 3D). The SNR decreased with increasing distance between the detection and illumination fiber. At a sampling rate of 30 ms, the SNR for the most distant collection fiber R5 was 8.03 and 5.93 for the calibration standard and tissue stack, respectively. This sampling rate was used for the remainder of the study.

### 3.3. Exploration of Tissue Constructs

The spectra were measured tissue constructs using collecting fibers labeled R1, R2, R3, R4, and R5 with distances from the center of the illumination fiber of 210, 345, 480, 615, and 750 µm, respectively. Figure 4A,B,D show example raw, normalized, and calibrated spectra, respectively, from constructs comprising the myocardium only (00000000), half myocardium and half aortic tissue (11110000), and aortic tissue only (11111111). The differences within the normalized and calibrated spectra were subtle. A spectrum from the calibration standard is shown in Figure 4C. The validation utilized a binary classification problem comprising constructs ‘00000000′ and ‘11111111′ using a CNN with a topology described in Table 1. We found that the CNN predicted with an accuracy of 98.82 ± 1.64% across all the fiber combinations and single fiber spectra.

A comparison of the spectra acquired from a construct of myocardial tissue only and the epicardial surface of a canine ventricle (unsectioned) is shown in Appendix A. The correlation of the two spectra is high (R_r_ = 0.9875).

### 3.4. Assessment of the Capabilities of the LSS System for Depth Detection

Using the spectra measured from the tissue constructs illustrated in Figure 2A, we trained the CNN for predicting the depth of a fibrotic tissue inset within the myocardium. The spectra used for training are illustrated in Appendix A. Figure 5A presents accuracies for the various fiber combinations from 10 trained CNNs. The use of the calibrated spectra resulted in insignificant differences between the mean accuracies of the networks when compared to those trained on the normalized, uncalibrated spectra. Therefore, we used normalized, uncalibrated spectra in the subsequent analyses. Figure 5B shows the mean and stddev of the difference between the spectra from the various constructs and the construct 11,111,111 (aortic tissue only). These differences were filtered to improve the visualization using a 1D Gaussian kernel with a stddev of 20 measures (~3.3 nm). Overall, the CNN performed well in the depth detection, achieving a high accuracy for the spectra from all of the combinations and single collection fibers. The networks achieved a maximal accuracy of 88.97 ± 2.49% with the R1R5 fiber combination. An accuracy of 86.64 ± 4.73% for R5 was the maximum for single fibers. Figure 5C,E show confusion matrices for the highest accuracy single fiber R5 and fiber combination R1R5. The lowest accuracy for a two-fiber combination was 63.19 ± 6.78% from R1R2 (Figure 5D). Generally, the fibers or combinations involving fibers distal, e.g., R4 or R5, from the illumination fiber generated more accurate depth predictions, and the combination of spectra led to higher accuracy than the spectra from their constituent single fibers.

### 3.5. Assessment of the Capabilities of the LSS System for Measuring Volume Fraction of Fibrotic Tissue

We next trained the CNN for the detection of volume fraction applying the spectra measured from the tissue constructs shown in Figure 2B. The spectra for training are illustrated in Appendix A. The accuracies are presented in Figure 6A. The differences between the spectra from the various constructs and the construct 11,111,111 are shown in Figure 6B. The R4R5 combination yielded the highest combination accuracy at 76.22 ± 1.98% (Figure 6E), and the highest accuracy of a single fiber was 76.33 ± 1.52% for R4 (Figure 6C). Often, misclassifications were to a similar arrangement. The least accurate single and combined fibers were R2 and R1R2 with respective accuracies of 48.94 ± 3.19% and 55.89 ± 4.98%. Figure 6D shows the confusion matrix for R1R2. The combinations involving two distal collection fibers (R3, R4, and R5) without the proximal fibers (R1 and R2) generally exhibited the highest accuracy.

### 3.6. Assessment of the Capabilities of the LSS System for Permutated Tissue Arrangements

Finally, we configured the tissue constructs as presented in Figure 2C to measure the spectra for training and testing of the CNN. The training spectra are shown in Appendix A. The accuracies for permutated tissue arrangements are presented in Figure 7A. Figure 7B presents the differences between the spectra from the various constructs and the construct 11111111. The highest single and combined fiber accuracies were 82.33 ± 0.97% and 84.25 ± 1.88% for R4 and R3R4, respectively. The confusion matrices for these networks are shown in Figure 7C,E. The lowest fiber combination accuracy was 75.83 ± 1.58% for the combination R1R3 (Figure 7D).

## 4. Discussion

In this study, we introduced a system for evaluating fibrosis in cardiac tissue using tissue constructs, LSS, and customized CNNs. The system was applied to evaluate the arrangements of collection fibers for classifying the tissue arrangements. An important finding of our studies was that the classifications based on concatenated spectra from two fibers were generally more accurate than the classifications utilizing the spectra from their respective constituent single collection fibers. For the detection of the fibrosis volume fraction of the constructs, the increase in average accuracy of two-fiber vs. single-fiber predictions was pronounced (7.21%). The increase in the average accuracy for two-fiber vs. single-fiber predictions was smaller for the detection of depth and permutated tissue arrangements at 3.13% and 2.59%, respectively.

Our approach was most accurate at identifying the differences in the depth of aortic insets and permutated tissue arrangements. It was less accurate in predicting the volume fraction. Often, the CNN misclassified to a similar tissue arrangement, e.g., a tissue construct differing by only a small number of tissue layers. These proximal predictions were accounted for by including them as partially (50%) correct predictions in our accuracy calculations (Equation (2)). Considering the clinical translation of the developed approach in applications such as fibrosis/scar mapping and the localization of conduction tissue, the implications of such classification errors are currently unexplored. We suggest that the impact of classification errors should be considered when establishing the training data for specific applications.

Our study builds on extensive prior work on LSS and optical properties of tissues. Several LSS systems with collection fibers at different distances to an illumination fiber have been introduced to increase the sensitivity proximal to the probe tip [6,24,25,26]. In some systems, e.g., [24,25], the illumination and collection fiber were integrally facilitated through a 50/50 beam splitter, and polarization filters were applied to isolate single scatter components in the detected spectra.

The spectra from distal single collection fibers and fiber combinations, including a distal fiber, typically produced a higher accuracy of predictions. Prior work on LSS described the total intensity of detected light as a function of reflection coefficients for multiply and singly scattered light, illumination intensity, and a transfer function of the optical system, including fibers [24]. Light detected from the fibers distal to the illumination fiber was assumed to be produced by multiple scattering as well as single scattering in the depth of the sample. Further work assuming multiple scattering revealed that the detection depth is dependent on the illumination-collection fiber spacing [27]. For an LSS system where the faces of the collection and illumination fibers are normal to the surface of a tissue, the modal line of the light distribution between these fibers follows a banana-shaped profile [4]. The depth of the modal line, z^max^, is maximal in the middle between the fibers and determined by their distance r_sd_:(3)$$z^{\max}\approx \frac{r_{\mathrm{sd}}}{2\sqrt{2}}$$

Thus, increased source-fiber separations increased the depth sensitivity, while smaller separations reduce the depth sensitivity. Our most proximal and distal collection fibers, R1 and R5, have a distance (center-center) to the illumination fiber of 210 and 750 µm, providing z^max^ of 74 and 265 µm, respectively. This analysis indicates that the combination of fibers provides complementary information, which could explain the improved accuracy of two-fiber vs. single-fiber classification.

Based on our study revealing the sensitivity at a depth of up to 1.6 mm, we suggest that light propagating deep into the tissue is a significant contributor to our measured spectra. In cardiac tissue, light in the visible spectrum (400–650 nm) has a transport mean free path length between 0.7 and 2.5 mm [28,29], while light with longer wavelengths (650–1000 nm) can have path lengths of up to 6 mm [30]. While our analyses are based on light detection within a partially different wavelength range (500–1100 nm), these path lengths point towards single-scattered light having a substantial influence on our measured spectra.

Visible inspection of the spectra indicated subtle, but characteristic features for specific tissue arrangements (Appendix A). However, simple classification approaches based on, e.g., slope extraction and thresholding did not yield promising classification results in our preliminary studies. Thus, for the classification of the spectra, we relied on CNNs, which were derived from conventional neural networks and are based on insights into vision processing in organisms. CNNs are thought to outperform conventional neural networks for many types of image and signal analyses. CNNs are known for their ability to extract and learn specific features in signals. Our studies suggest that CNNs are efficient and reliable for the classification of spectra.

An alternative approach for the analysis of the spectra is simulating light transport in tissue models and optimizing these models by the comparison of the results from the simulation and the measurements. Well established are the Monte Carlo methods for these simulations [31,32]. However, computational demands of the Monte Carlo methods are a major limiting factor. Due to the computational demands, it appears difficult to develop this approach for real-time classification of spectra.

Our LSS system measured the single or dual spectra at a rate of 33 Hz. This is well under the 200 Hz upper limit of the spectrometer used in this study. We chose 33 Hz to assure a high SNR for all of the collection fibers. The current classification of the spectra is a-posteriori. However, we suggest that real-time visualization of tissue classifications using CNNs is feasible and will provide a clinician with important information for intraoperative decision-making.

Our exploration was based on varying the location and number of aortic insets in a myocardial tissue construct. Due to its higher collagen content and density of nuclei, aortic tissue has a higher scattering coefficient than the myocardium [22]. These differences in the scattering coefficient explain the differences in spectral intensities in Figure 3A and Figure 4A. However, the intensity differences between the tissues were removed by normalization and not considered by the CNN. An advantage of the normalization is that the classification is not based on the absolute spectral intensities, which expectedly will vary dependent on, e.g., intensity of the light source for illumination, coupling of the collection fibers, and the spectrometers, as well as the length and bending of the optical fibers for both illumination and collection. In particular, for catheterized applications in the atria, we expect highly variable bending of the fibers, and thus effects on the illumination intensity and light collection.

We utilized the collagen-rich aortic wall as a model of compact fibrosis and scar tissue, as well as connective tissue found, e.g., in nodal regions of the heart. An experimental design was aimed at an approximation of tissue distributions in the normal and diseased heart. Our studies on the depth and volume fraction of an aortic tissue inset constitute a basis for the development of a system for detection and mapping of atrial and ventricular fibrosis. In animal models and patients, fibrosis is a hallmark of several cardiac pathologies [33,34,35,36]. We note that fibrosis has many phenotypes. For example, fibrosis can be interstitial, compact, diffuse, or patchy [37]. Additionally, the non-cellular and cellular composition of fibrosis is highly variable. We suggest that our study provides a framework for the identification of the phenotype of fibrosis and its composition. The application of the framework will require an assessment of the fibrosis composition, e.g., using confocal microscopy, and applying this assessment in the training as outlined in our study.

Furthermore, our studies on the volume fraction of aortic tissue could guide the development of a system for the detection of scars, e.g., caused by radiofrequency ablation and cryoablation. Clinically, the presence of a scar is an indicator of successful ablation therapy for a cardiac arrhythmia, such as atrial fibrillation [38,39]. We note, however, important differences in the composition of the cardiac tissues during and acutely after ablation versus the aortic tissue applied. The assessment of the developed approach on ablated tissues will be a crucial step in such a development. One application of our studies on the permutated arrangements is the detection of conduction tissues, e.g., the sinoatrial and atrioventricular node, which are embedded in connective tissues. For all of these applications, the depth of detection is crucial.

We acknowledge that the presented study has several limitations. Firstly, we applied fixed excised tissues as opposed to actively contracting and passively deformed tissues in the heart. The validation of the approach, similarly as in our prior work on fibrosis detection in the beating in situ heart of a canine model, will be an important step towards translation of the approach [13]. A further limitation is related to the assembly of the sections of the myocardium and aortic tissue to generate the tissue constructs. Our initial studies suggested that the sectioning and assembly had only minor effects on the spectra (Appendix A), but we cannot exclude that our approach alters the optical properties of the tissues to some degree. While allowing for the precise control of the degree and distribution of tissue composition, the constructs are artificial and only partially resemble the compositions of the various types of cardiac fibrosis or scar that have been found in animal and patient studies. Our tissue constructs resemble most closely the myocardium with compact fibrosis or an ablation scar. The limitations are also associated with the specific machine learning approach applied in our work. We investigated only CNNs. While we extensively explored CNN topologies and the parameters of network layers, we cannot exclude that other approaches or further optimization would yield a higher training efficiency and classification accuracy. Other machine learning approaches potentially suitable for the classification of spectra include k-means clustering and fully connected neural networks. A limitation of our study is that we did not attempt to optimize the range and sampling of the spectral wavelengths with respect to the classification accuracy. We used light in the visible and near-infrared wavelength range. We anticipate that the optimization of wavelength ranges and their sampling will improve the computational efficiency and provide design input for LSS systems for specific applications. We did not apply calibrated spectra for the CNN-based training and testing. While differences in the average classification accuracy for the calibrated and non-calibrated spectra were not significant, a concern is that differences in spectrometers are reflected in the trained network. This complicates the sharing of networks trained on one LSS system and their usage on another system. Lastly, we explored only the arrangements where one or two collection fibers provided input for the CNN. A larger number of collection fibers might yield higher accuracies but would increase the complexity of the LSS system.

## 5. Conclusions

In conclusion, we introduced an approach based on LSS and CNNs for the optical characterization of cardiac tissue. The approach was explored with tissue constructs with a thickness of 1.6 mm. The approach achieved high accuracies for detecting the depth of a fibrotic tissue inset, volume fraction of fibrotic tissue, and permutated tissue arrangements. We suggest that the integration of the approach with catheter technology will yield a novel tool in cardiology and heart surgery. The potential applications include catheterized fibrosis quantification and mapping in the beating heart in situ, lesion assessment during ablation therapy, and the detection of conduction tissue during cardiac surgery.

## 6. Patents

The University of Utah has submitted a provisional patent application associated with the techniques described in this paper.

## Figures and Tables

**Figure 1 sensors-21-06033-f001:**
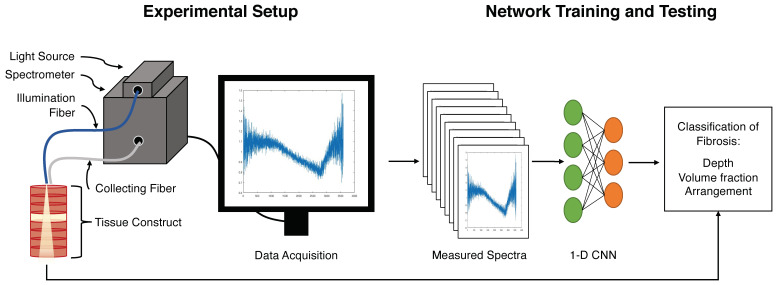
Experimental setup. The setup comprised a stabilized tungsten–halogen light source and a Czerny–Turner type charge-coupled device (CCD) spectrometer. Raw spectra from cardiac tissues and tissue constructs were collected and preprocessed. A portion of the processed spectra was used to train a CNN. The remaining data were used to evaluate the performance of the CNN for classification of tissue.

**Figure 2 sensors-21-06033-f002:**
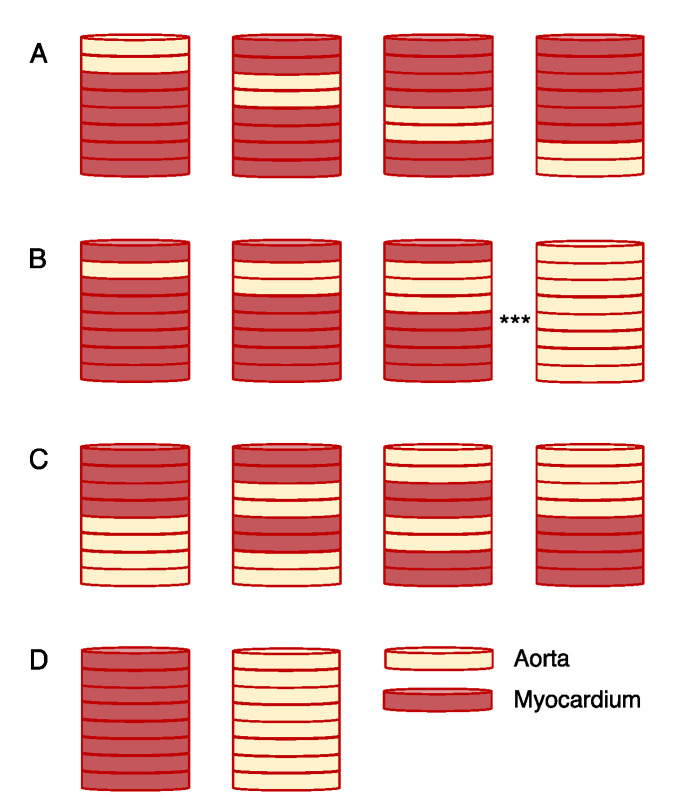
Tissue constructs. (**A**) For studies on depth detection of insets of fibrotic tissue, two 200 µm thick sections of aortic tissue were placed at various locations within the 1.6 mm thick constructs. (**B**) Tissue constructs for analysis of LSS sensitivity for volume fractions. The sensitivity was evaluated by successively increasing the ratio of aortic tissue to the myocardium. (**C**) Permutated tissue section arrangements were used to assess the sensitivity of LSS to tissue heterogeneities. (**D**) Constructs comprising only myocardial and aortic tissue sections complemented the constructs in (**A**–**C**) in all the analyses.

**Figure 3 sensors-21-06033-f003:**
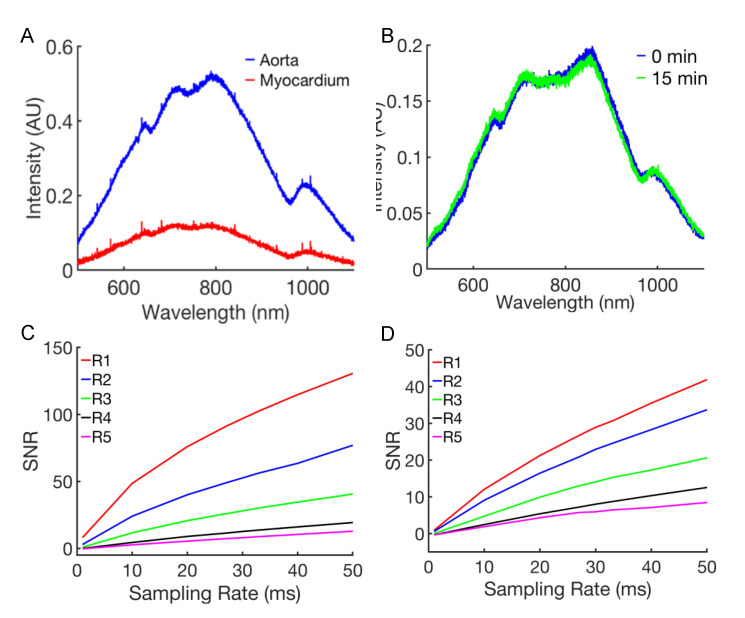
Spectroscopy of the aorta and ventricular myocardium from a rat. (**A**) Raw reflectance spectra from live aorta and live ventricular myocardium. The spectra were pronouncedly different. In particular, the spectrum from the myocardium exhibited smaller intensities than the spectrum from the aorta. (**B**) Differences between raw spectra from the myocardium before and after 15 min of fixation with paraformaldehyde were marginal. (**C**,**D**) Signal-to-noise ratio (SNR) for collection fibers R1 to R5 and various sampling rates measured using the (**C**) calibration standard and (**D**) fixed ventricular tissue.

**Figure 4 sensors-21-06033-f004:**
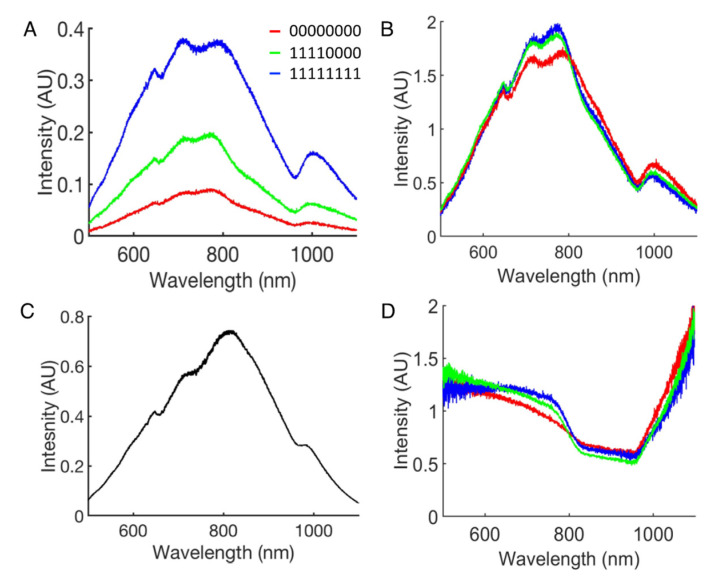
Processing of spectra measured from tissue constructs with different ratios of myocardium to aortic tissue. (**A**) Raw spectra from construction of 100% myocardium, 50% myocardium/50 aortic tissue, and 100% aortic tissue. (**B**) Spectra from (**A**) after normalization. (**C**) Spectrum measured from calibration standard. (**D**) Spectra from (**B**) after calibration using the spectrum from (**C**).

**Figure 5 sensors-21-06033-f005:**
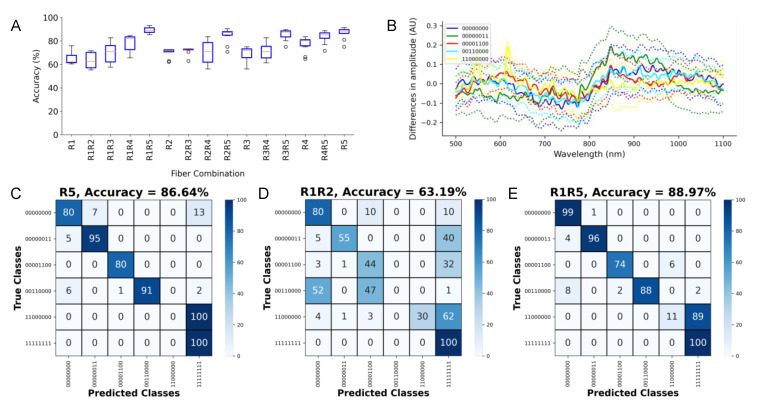
Assessment of depth detection of fibrotic insets. (**A**) Accuracy for single and combinations of collection fibers. (**B**) Mean (bold line) and stddev (dotted line) of the difference between spectra from the various constructs and the construct 11111111 (aortic tissue only). Confusion matrix for the (**C**) highest accuracy single fiber R5, (**D**) lowest accuracy fiber combination R1R2, and (**E**) highest accuracy fiber combination R1R5. In the class name, “1” denotes aortic tissue, while “0” denotes the myocardium.

**Figure 6 sensors-21-06033-f006:**
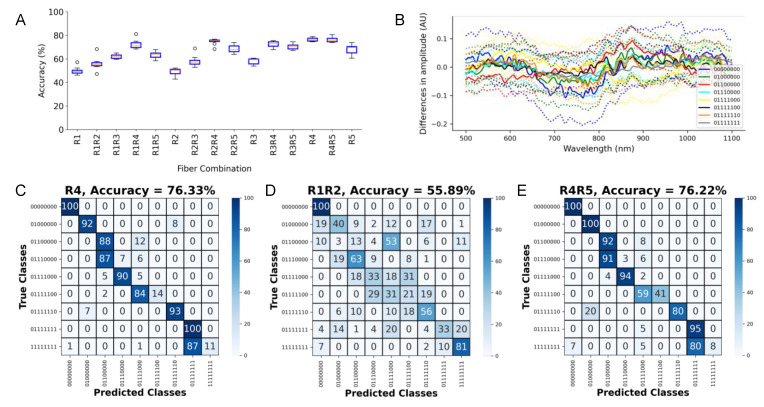
Assessment of quantification of volume fraction of fibrosis. (**A**) Accuracy for single and combinations of collection fibers. (**B**) Mean (bold line) and stddev (dotted line) of the difference between the spectra from the various constructs and the construct 11111111 (aortic tissue only). Confusion matrix for the (**C**) highest accuracy single fiber R4 fiber, (**D**) lowest accuracy fiber combination R1R2, and (**E**) highest accuracy fiber combination R4R5. In the class name, “1” denotes aortic tissue, while “0” denotes the myocardium.

**Figure 7 sensors-21-06033-f007:**
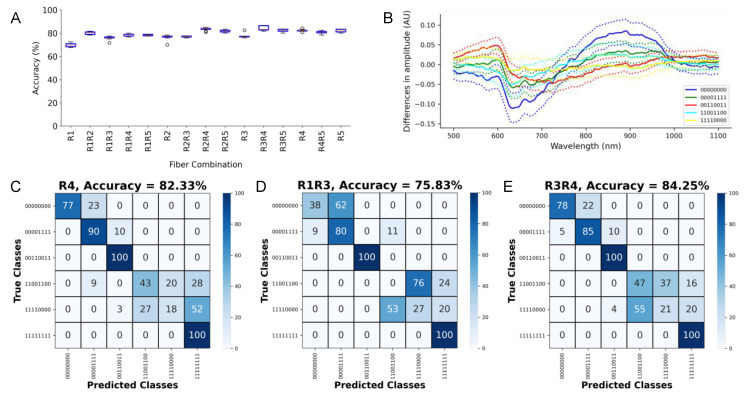
Assessment of detection of permutated tissue arrangements. (**A**) Accuracy for single and combinations of collection fibers. (**B**) Mean (bold line) and stddev (dotted line) of the difference between the spectra from the various constructs and the construct 11111111 (aortic tissue only). Confusion matrix for the (**C**) highest accuracy single fiber R4, (**D**) lowest accuracy fiber combination R1R3, and (**E**) highest accuracy fiber combination R3R4. In the class name, “1” denotes aortic tissue, while “0” denotes the myocardium.

**Table 1 sensors-21-06033-t001:** Convolutional Neural Network Configuration.

Level	Layer Type	Parameters
1,4,7	Convolution	Filter numbers: 8, 10, and 12 respectively, kernel size: 5, stride: 1 First layer input size: 3587 for single spectra or 7174 for combined spectra, mean normalization
2,5,8	ReLU	Rectified linear unit activation layer
3,6,9	Max Pooling	Pool size: 2, stride: 0, padding: None
10	Softmax/Sigmoid	Exponential activation layer for multi-class and binary classification respectively

## Data Availability

Spectra and associated data from this study are stored at The Hive: University of Utah Research Data Repository, doi:10.7278/S50D-3Q4J-SC4Y.

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
