# Peer review of "Towards Intraoperative Quantification of Atrial Fibrosis Using Light-Scattering Spectroscopy and Convolutional Neural Networks"

_sensors, 2021, doi:10.3390/s21186033_

Round 1

Reviewer 1 Report

In the manuscript “Towards Intraoperative Quantification of Atrial Fibrosis Using Light Scattering Spectroscopy and Convolutional Neural Networks” Knighton et al. apply light scattering spectroscopy (LSS) in combination with machine learning to characterize the composition of cardiac tissues. By using a 200 micron diameter illumination fiber and five 105- micron-diameter collection fibers with approximately 100 micron increments in the source-detector separations the authors capitalize on the LSS ability to obtain depth specific spectroscopic information. While the majority of the LSS studies till now employ physical models of light scattering in tissue to associate measured spectra with the tissue morphological and biochemical characteristics, this paper appears to be the first where the physical tissue model is replaced with the machine learning approach. Using this approach the authors appear to be able to predict the depth placement of the fibrotic tissue within myocardium. The authors provided extensive referencing of publications relevant to the presented work. One suggestion would be to also reference the paper that started the LSS field: Perelman, L.T.; Backman, V; Wallace, M.; Zonios, G.; Manoharan, R.; Nusrat, A.; Shields, S.; Seiler M.; Lima, C.; Hamano, T.; et al. Observation of periodic fine structure in reflectance from biological tissue: a new technique for measuring nuclear size distribution. Physical Review Letters 1998, 80, 627-630. doi: 10.1103/PhysRevLett.80.627. Overall this is an interesting paper which presents a somewhat new twist on how LSS can be applied to characterize tissue which would be of interest to the general biomedical optics community. In my opinion the manuscript should be published in Sensors.

Author Response

Response: We appreciate the positive assessment of our manuscript. We added the reference to the Perelman et al paper as suggested. Indeed, this is a central paper on light scattering spectroscopy in biological tissues that constitutes a fundament for our work.

Reviewer 2 Report

This is a very good article that can of course be published.  Nevertheless, I would like to get some clarifications on Figures 3 and 4.  

Does the mathematical analysis of these spectra make sense, namely, from the decomposition into Gaussian and Lorentzian components?

How does the quality of the samples affect the reflectance spectra? Is the limitation of the measurements on the short-wavelength side (360 nm) of the spectrum fundamental or purely technical? What can be expected if measurements are extended to 200-250 nm?

Author Response

This is a very good article that can of course be published.  Nevertheless, I would like to get some clarifications on Figures 3 and 4.  

Response: We appreciate the positive feedback. We significantly expanded the captions of Fig. 3 and 4 to provide more detail on the measurements of spectra and their processing.

Does the mathematical analysis of these spectra make sense, namely, from the decomposition into Gaussian and Lorentzian components?

Response: Gaussian and Lorentzian decomposition is indeed an established and insightful approach for the spectroscopic assessment of materials. However, such a decomposition is difficult for highly heterogeneous materials with many constituents such as cardiac tissues. This heterogeneity and the diversity of scatterers are reflected in the calibrated spectra in Fig. 4D. Due to this complexity, we applied a machine learning approach, which does not require a decomposition.

How does the quality of the samples affect the reflectance spectra?

Response: Sample quality, in particular, the quality of the cardiac tissues and tissue constructs, was an important consideration in our studies. We implemented reliable protocols for fixation, tissue preservation, and sectioning. We also were very careful with the assembly of the tissue constructs to avoid air pockets between the tissue sections. We did not produce and study low-quality samples, but would expect that spectra differ if e.g. the tissue microstructure is not well preserved or sectioning damaged the slices.  

Is the limitation of the measurements on the short-wavelength side (360 nm) of the spectrum fundamental or purely technical? What can be expected if measurements are extended to 200-250 nm?

Response: Primarily for technical and manufacturing reasons, we limited our investigations to the visible and NIR range. Conventional light sources and sensors commonly cover this range. A concern with the UV range is also that absorption is much higher for many tissue constituents. This would limit depth (see e.g. Fig. 5, Huang et al, Catheter-based optical approaches for cardiovascular medicine: progress, challenges and new directions. Progress in Biomedical Engineering 2020, 2, 032001) crucial for many cardiac applications.